# Application of Functional Data Analysis to Identify Patterns of Malaria Incidence, to Guide Targeted Control Strategies

**DOI:** 10.3390/ijerph17114168

**Published:** 2020-06-11

**Authors:** Sokhna Dieng, Pierre Michel, Abdoulaye Guindo, Kankoe Sallah, El-Hadj Ba, Badara Cissé, Maria Patrizia Carrieri, Cheikh Sokhna, Paul Milligan, Jean Gaudart

**Affiliations:** 1Sciences Economiques et Sociales de la Santé et Traitement de de l’Information Médicale (SESSTIM), Institut de Recherche pour le Développement (IRD), Institut National de la Santé et de la Recherche médicale (INSERM), Aix Marseille Université, 13005 Marseille, France; aguindo15@yahoo.fr (A.G.); levisallah@gmail.com (K.S.); pmcarrieri@aol.com (M.P.C.); 2Aix Marseille School of Economics (AMSE), Centrale Marseille, Ecoles des Hautes Etudes en Sciences Sociales (EHESS), Centre National de la Recherche Scientifique (CNRS), Aix Marseille Université, 13001 Marseille, France; pierre.michel@univ-amu.fr; 3Mère et Enfant face aux Infections Tropicales (MERIT), Institut de Recherche pour le Développement (IRD), Université Paris 5, 75006 Paris, France; 4Unité de Recherche Clinique Paris Nord Val de Seine (PNVS), Hôpital Bichat, Assistance Publique—Hôpitaux de Paris (AP-HP), 75018 Paris, France; 5Unité Mixte de Recherche (UMR), Vecteurs-Infections Tropicales et Méditerranéennes (VITROME), Campus International Institut de Recherche pour le Développement-Université Cheikh Anta Diop (IRD-UCAD) de l’IRD, Dakar CP 18524, Senegal; el-hadj.ba@ird.fr (E.-H.B.); cheikh.sokhna@ird.fr (C.S.); 6Institut de Recherche en Santé, de Surveillance Épidémiologique et de Formation (IRESSEF) Diamniadio, Dakar BP 7325, Senegal; badara.cisse@iressef.org; 7London School of Hygiene and Tropical Medicine, London WC1E 7HT, UK; Paul.Milligan@lshtm.ac.uk; 8Aix Marseille Université, Assistance Publique—Hôpitaux de Marseille(APHM), INSERM, IRD, SESSTIM, Hop Timone, BioSTIC, Biostatistic and ICT, 13005 Marseille, France; jean.gaudart@univ-amu.fr

**Keywords:** functional data analysis, time series clustering, malaria patterns, malaria dynamic

## Abstract

We introduce an approach based on functional data analysis to identify patterns of malaria incidence to guide effective targeting of malaria control in a seasonal transmission area. Using functional data method, a smooth function (functional data or curve) was fitted from the time series of observed malaria incidence for each of 575 villages in west-central Senegal from 2008 to 2012. These 575 smooth functions were classified using hierarchical clustering (Ward’s method), and several different dissimilarity measures. Validity indices were used to determine the number of distinct temporal patterns of malaria incidence. Epidemiological indicators characterizing the resulting malaria incidence patterns were determined from the velocity and acceleration of their incidences over time. We identified three distinct patterns of malaria incidence: high-, intermediate-, and low-incidence patterns in respectively 2% (12/575), 17% (97/575), and 81% (466/575) of villages. Epidemiological indicators characterizing the fluctuations in malaria incidence showed that seasonal outbreaks started later, and ended earlier, in the low-incidence pattern. Functional data analysis can be used to identify patterns of malaria incidence, by considering their temporal dynamics. Epidemiological indicators derived from their velocities and accelerations, may guide to target control measures according to patterns.

## 1. Introduction

The development of technology has increasingly enabled the use of sophisticated tools to collect and store large amounts of complex data, particularly in scientific fields. These data are often continuous but observed over a finite number of points (discretization points) [1,2,3]. This is the case for meteorological data, electrocardiogram, time series, growth curves, for example.

A functional data approach would be better adapted to handle these data by taking into account some of their particularities. Indeed, this approach is useful to handle a large sample of spatial units (villages) allowing comparison between them and to reduce data dimensions (number of observations) for long time series. In addition, the number of observations may be higher than the size of the sample making statistical analysis difficult. The observations are not always made at a regular time lag (every hour, every day etc.) and this latter may differ from one place to another [1,3]. Moreover, the use of functional data also allows the estimation of the velocity and acceleration of the time series.

As a result, a considerable amount of research has been dedicated to the development of statistical methods and tools for analysis of functional data [1,2,4,5,6]. The works by Ramsay et al. have made these approaches popular, and R and MATLAB programs (The R Foundation for Statistical Computing, Vienna, Austria) have made the methods available to a wider group of researchers [5]. Applications in public health and biomedical sciences have been reviewed by Ullah and Finch (2013) [7].

In areas with low malaria transmission, because of the spatial heterogeneity of malaria incidence, World Health Organization (WHO) recommends the development of targeted control strategies adapted to the local epidemiological context [8]. Effective targeting requires identification of transmission foci or hotspots based on epidemiological data. Existing approaches used, to target malaria risk areas are based on aggregated incidence or prevalence rate, [9,10,11,12,13,14,15] in large discrete time sub-periods [16,17,18,19,20]. Thus, malaria risk areas were identified every rainy season or every year or another large sub-period and sometimes the status at malaria risk of areas between sub-periods can change. These approaches do not provide information about the trend or temporal dynamic of malaria and continuous time approaches are useful for dynamic analysis.

Using the functional data approach, the observed malaria incidences can be described by estimated smooth functions (curves) in order to understand the underlying temporal trends of malaria. These smooth functions can be obtained for each of a large number of spatial units (villages), clustering algorithms can then be used to identify broad types of temporal patterns according to the characteristics of their dynamics (temporal trends). This would help to guide the development and implementation of targeted control strategies in the local context.

In addition, for further understanding of malaria incidence dynamics, the velocity and the acceleration (velocity variation) are useful. Indeed, the velocity is the first derivative function which gives information over time about when the malaria incidence increases (growth phase period) or decreases (decline phase period). The acceleration, i.e., the variation of epidemic speed (velocity) is the second derivative function. This indicates how malaria incidence increases or decreases over time: quickly or slowly [21,22]. Thus, temporal variations of velocity and acceleration together provide information about the malaria dynamic. Moreover, key features of the malaria dynamic derived from velocity and acceleration functions as onsets, peaks, ends, and their lags between patterns are useful to refine targeted intervention schedules.

In this paper, we introduced an approach based on functional data analysis to identify patterns of malaria incidence over a five-year period at village scale, in west-central Senegal. In addition, with the epidemiological indicators determined from the velocity and the acceleration of the resulting patterns, we investigated the spatiotemporal variation and features of malaria incidence in local context, in order to guide the targeted malaria control measures in a low transmission area and local context.

## 2. Methods

### 2.1. Study Area and Dataset

The data used for this study were collected between January 2008 and December 2012 during a field trial of Seasonal Malaria Chemoprevention (SMC) among children from 575 villages in west-central Senegal [23,24]. This area is a part of the two national rural health districts, Bambey and Fatick, where the national malaria control program estimated the incidence under 5 cases/1000 person-years in 2018 [25]. The protocols for the field studies were approved by Senegal’s Conseil National pour la Recherche en Santé and the ethics committee of the London School of Hygiene and Tropical Medicine. The SMC trial [23] was registered number NCT00712374. The datasets analyzed during the current study are available from the corresponding author on reasonable request.

Malaria surveillance was maintained in 38 health facilities serving a population of about 500,000 living in 575 villages (single villages or groups of adjacent hamlets). Malaria cases were patients examined at health facilities with fever or history of fever, in the absence of evident alternative causes of the fever, who had a positive rapid diagnostic test (RDT). Date of diagnosis, village of residence, and other details were obtained for each case from facility registers. The surveillance system is described by Cissé et al. [23]. The population was counted through a census in 2008 and updated through visits to each household at approximately 10 months intervals from 2008 to 2012. The coordinates of each village center were obtained by Global Positioning System (GPS).

For this analysis only aggregated data by villages were used.

### 2.2. Statistical Analysis

Our approach consisted of three stages.

In the first stage, we estimated malaria incidence curves (smooth functions or functional data) from these 575 villages (from January 2008 to December 2012) using a basis functions representation (village-level temporal trends). 

In the second stage, hierarchical clustering was applied to the malaria incidence curves for classifying villages with similar temporal trends together. To obtain an optimal classification, several dissimilarity measures and validity indices were used.

In the third stage, resulting patterns were characterized using predefined epidemiological indicators from their velocity and acceleration functions to describe the overall features of temporal patterns identified.

#### 2.2.1. Estimating the Smooth Function (Functional Data) for Each Time Series 

The time series of the observed weekly malaria incidence (i.e., the number of confirmed cases per week divided by the total population of the village at this week) was determined for each of the 575 villages. A square root transformation was applied to these incidence rates to stabilize the variance [4]. The functional data method [4,5] states that the square root of observed malaria incidence rate for a village *i* at a week *j* is the sum of a function on the time continuous of week *j* and an error term (1).
(1)$$y_{ij}=\sqrt{Inc_{ij}}=x_{i}\left(t_{j}\right)+\varepsilon_{ij}\;i=1,\ldots ,\;575,\;\;j=1,\ldots ,\;261$$
where $x_{i}$ is a regular (smooth) function which describes the temporal pattern of malaria incidence in village *i*, *t_j_* is the continuous time of week *j*, and $\varepsilon_{ij}$ is an error term representing the difference between the function value and the observed data for village *i* at week *j*.

The function $x_{i}$ is approximated by a finite sum of linear combination of basis functions (2):(2)$$x_{i}\left(t\right)=\overset{K}{\underset{k=1}{\sum }}c_{ik}\;\varphi_{k}\left(t\right)$$
where $\varphi_{k}$ are basis functions, *K* represents the total number of basis functions and $c_{ik}$ are the coefficients obtained by the least squares method by minimizing the penalized error sum of squares (3) after replacing in (3) the formula of $x_{i}\left(t_{j}\right)$ by Equation (2):(3)$$SSE\left(x_{i}\right)=\overset{T}{\underset{j=1}{\sum }}(y_{ij}-x_{i}\left(t_{j}\right)){\;}^{2}+\lambda \int_{P}{\left|x_{i}^{\prime \prime }\left(t\right)\right|}^{2}\;dt,\;\;i=1,\;\ldots ,\;575$$
where λ is the non-negative smoothing parameter, *P* the studied period expressed here as [1, 261] in which times are continuous and $x_{i}^{\prime \prime }$ the second derivative function with
(4)$$\int_{P}{\left|x_{i}^{\prime \prime }\left(t\right)\right|}^{2}\;dt<\infty$$

The penalty term (4) controls the smoothness of the estimate for $x_{i}\left(t\right)$. Large values of lambda λ yield nearly linear curve estimates while small values of lambda yield wiggly curve estimates getting closer to observed data.

To estimate the underlying smooth function or functional data $x_{i}$, the family of basis functions $\varphi_{k}$, their total number *K* and the smoothing parameter λ should be chosen. The basis functions are families of known functions [4,5]. Several basis functions are possible (B-spline, Fourier, exponential etc.), but they have to be chosen according to the nature of the data. In this work, we used cubic B-splines to avoid periodic smoothing [4,5,7]. Indeed, even if malaria incidence has a periodic nature, the level of intensity of incidence is not the same over seasons, in contrast with Fourier basis functions, which will show the same level intensity as that of seasonal incidence.

While the choice of the smoothing parameter is very important, there is no universal rule for an optimal choice. However, a number of criteria are available, including the generalized cross-validation (GCV), which we used in this study [26].

#### 2.2.2. Dissimilarity Measures and Hierarchical Ascending Clustering on Smooth Functions 

Hierarchical ascending clustering [27] is one of the most popular unsupervised clustering algorithms grouping similar elements such that the elements in the same group are more similar to each other than the elements in the other groups. At the beginning, each element is a cluster, then elements are grouped according to a dissimilarity measure and aggregation criteria until having one cluster grouping all elements. The advantage with this clustering method is its ability to work without prior number of clusters, which can influence the results compared to the K-means method for example. Another advantage is its dendrogram tool showing different cluster possibilities.

To perform a hierarchical ascending clustering on smooth functions (functional data or curves), a dissimilarity measure between them was necessary to assess their proximity before each grouping step. In the context of time series, several dissimilarity measures are proposed in the literature [28,29]. This work focused on those based only on the data value or level intensity, and those based, in addition, on the temporal evolution or behavior of data over time that would adapt to the functional data [26,28,29,30]. 

For those based only on data values, we have selected four: the Euclidean distance [29] ($d_{EUC}$) based on the point-to-point differences between observations of the two curves, the Lp-metric [26] estimating the surface between two functional data (curves) ($d_{FDA}$), the dynamic time warping [31] ($d_{DTW}$) providing a measure of distance insensitive to local compression, stretching, and the optimal deformation of one of the two curves compared to the other, and the discrete wavelet transformation [29] ($d_{DWT}$) measuring the dissimilarity between the wavelet approximations associated with the observations of curves. 

For those based on data values and behavior [28,30] ($d_{CORT}$) (5), a temporal correlation (6) between two functional data (curves) was combined with each of the dissimilarity measures based only on data values. The contributions of the data value part and the temporal correlation part were adjusted by an adaptative function according to a value of a given non-negative parameter ξ (Table 1). Table 1 comes from the original article [30], which developed the $d_{CORT}$ dissimilarity measure. So a dissimilarity measure based on data values and behavior was obtained by combing one of the four measures based only on data value ($d_{EUC}$, $d_{FDA}$, $d_{DTW}$, $d_{DWT}$) and the temporal correlation (6) according for each value of the parameter ξ in Table 1. 

Thus, a total of 20 dissimilarity measures were used in this analysis. For example, $d_{EUCCORT1}$ is the dissimilarity measure based on temporal correlation (CORT) and the Euclidean distance ($d_{EUC}$) (Equation (5)) when ξ is equal to 1 (Table 1), this considered 46.2% of behavior contribution (given by CORT) and 53.7% of values contribution (given by $d_{EUC}$). The others dissimilarity measures were obtained in the same way, and when ξ = 0 (Table 1), we had the four dissimilarity measures based only on values.
(5)$$d_{CORT}\left(x_{i},x_{i^{\prime }}\right)=f_{\xi }\left[CORT\left(x_{i},x_{i^{\prime }}\right)\right]*d\left(x_{i},x_{i^{\prime }}\right)$$
(6)$$CORT\left(x_{i},x_{i^{\prime }}\right)=\frac{\sum_{t=1}^{T-1}(x_{i}\left(t+1\right)-x_{i}\left(t\right))(x_{i^{\prime }}\left(t+1\right)-x_{i^{\prime }}\left(t\right))}{\sqrt{\sum_{t=1}^{T-1}{\left(x_{i}\left(t+1\right)-x_{i}\left(t\right)\right)}^{2}}\sqrt{\sum_{t=1}^{T-1}{\left(x_{i^{\prime }}\left(t+1\right)-x_{i^{\prime }}\left(t\right)\right)}^{2}}}$$

The adaptative function [30] $f_{\xi }\left(u\right)=\frac{2}{1+\exp\left(\xi u\right)},\;\;\xi \geq 0$ is used to adjust the percentage of contribution of value and behavior according to the value of the parameter ξ. 

At this stage only the Euclidean distance and the dynamic time warping distance were implemented in R package with this dissimilarity measure. We have written a R program to estimate the functional Euclidean distance and the discrete wavelet transformation dissimilarity based on valued and behavior with the same formula above. 

Thus, hierarchical ascending clustering (HAC) was performed on smooth functions using each dissimilarity measure with Ward aggregation method [32]. This was to find the most able dissimilarity measure to assess the difference between curves for reaching a better quality of classification. Thus, we obtained 20 HAC results. To assess the HAC results according to potential numbers of patterns (chosen after examination of dendrograms), four validity indices were used in a multidimensional space for functional data [33,34,35]: connectivity, Dunn, silhouette width, and the percentage of inertia explained by the number of patterns R^2^. The connectivity indicates the degree of connectedness of the clusters, as determined by the *k*-nearest neighbors (in this work *k* = 10). The connectivity has a value between 0 and infinity and should be minimized. Both the silhouette width and the Dunn index combine measures of compactness and separation of the clusters. The silhouette width is the average of each curve’s silhouette value. The silhouette value measures the degree of confidence in a particular clustering assignment and lies in the interval [−1, 1] with well-clustered curves having values near 1 and poorly clustered curves having values near −1. The Dunn index is the ratio between the smallest distance between curves not in the same cluster to the largest intra-cluster distance. It has a value between 0 and infinity and should be maximized. To choose the final or optimal number of malaria incidence patterns, we performed a principal component analysis (PCA) [36] on assessed HAC results to look for the one which showed the best criteria of validity indices, i.e., with connectivity index close to 0, high Dunn index, and silhouette width and R^2^ close to 1. 

In these two first steps, we worked on the functional data of the square root transformation of the observed time series, but for the following step, we applied square transformations to obtain the functional data corresponding to the observed time series, on which interpretations where based.

Let *Q* be the number of patterns identified by the HAC, we defined the functional data of each pattern by $C_{q}\left(t\right)$, *q* = 1, …, *Q* being the cumulative weekly incidence of the villages belonging to the pattern *q*. Then, their 95% point-wise confidence intervals were computed by adding and subtracting two of the standard errors, that is, the square root of the sampling variances, to the actual fit [4].

#### 2.2.3. Velocity and Acceleration 

To further describe the malaria incidence patterns, the first (velocity) and second (acceleration) derivative were determined for each functional data of a pattern. Their variations over time indicated the growth and decline phase periods in each pattern, and the degree of speed: quickly or slowly. Thus, with mathematical properties of univariate function optimization [21] and one-dimensional kinematics in physics [22], seven epidemiological indicators based on velocity and acceleration were defined (Figure 1, Table 2). These epidemiological indicators were as follows: the beginning of seasonal outbreaks and the start acceleration of the growth phase (A); the beginning of the pre-slowdown of the growth phase (B); the deceleration’s beginning of growth phase (C); the peak (D) also corresponding after to the beginning of the acceleration of the decrease phase; the beginning of the deceleration of the decrease phase (E); the beginning of the tail (F); the end of the seasonal outbreaks (G). Finally, a PCA was performed on the durations: AB, AD, CE, DG, FG, BF, and AG to look for those that characterized patterns of seasonal outbreaks.

Statistical analyses were performed with R^®^ software (The R Foundation for Statistical Computing, Vienna, Austria) R 3.4.2 version. Maps were produced using QGIS^®^ software (Open Source Geospatial Foundation, Boston, MH, USA) QGIS 3.10.1 version.

## 3. Results

### 3.1. From Observed to Smoothed Malaria Incidence 

The observed time series of malaria incidence for each of the 575 villages from January 2008 to December 2012 were determined (Figure 2, Panel A). The observed malaria incidence ranged from 0 to 183 cases/1000 person-years at the village level, and the median was 4 cases/1000 person-years with interquartile range (2, 9). At village and week levels, the observed malaria incidence ranged from 0 to 17,000 cases/100,000 person-weeks (Figure 2, Panel A).

Because of the high variability, the transformation with square root function was applied to the time series of Figure 2, Panel A to obtain the observed time series of malaria incidence in square root scale (Panel B) as explained in the Methods section Equation (1).

With the transformed observed time series (Panel B), the search for the optimal number of basis functions and the optimal smoothing parameter gave $K_{opt}=110$ and $\lambda_{opt}=103,$ which minimized the error by GCV equal to 11.8 with a standard deviation of σ=0.12. Using these optimal parameters, the smoothed transformed time series of malaria incidence for each village were determined (Figure 2, Panel C). 

For epidemiological interpretation, we applied the square function (reciprocal function) to obtain the smoothed time series of malaria incidence corresponding to the observed time series (Figure 2, Panel D).

### 3.2. Identification of Malaria Incidence Patterns

Three patterns with the DTWCORT1 dissimilarity measure (3DTWCORT1 HAC result) were obtained with the application of HAC on the smoothed transformed times series (Figure 2, Panel C). Indeed, these patterns were chosen based on the PCA performed on assessed validity indices across the HAC results obtained with each of the 20 dissimilarity measures for 3 and 4 number patterns (Appendix A, Table A1, Figure A1). 

In addition, the dimension 1 represented high Dunn index and silhouettes, and low connectivity (Figure 3, Panel A). The dimension 2 essentially represented the percentage of inertia explained by the patterns R^2^ (Figure 3, Panel A). The best classification should therefore be located in the upper right of the factorial plane of the dissimilarity measures and the number of patterns (Figure 3, Panel B). The DTWCORT1 dissimilarity measure took into account 46.2% of the temporal correlation between functional data and 53.7% of the geometric distance.

The high-incidence pattern (high pattern) consisted of a set of 12 villages with the highest observed average incidence over the five-year study period (114 cases/1000 person-years), mainly located in the southern part of the study area (Figure 4). Its smoothed seasonal outbreaks peaks ranged from 227 (95% CI: [65, 487]) to 884 cases/100,000 person-weeks (95% CI: [420, 1518]) (Figure 5, Table 3). 

The intermediate-incidence pattern (intermediate pattern) included 97 villages had 13 cases/1000 person-years as observed average incidence over the study period, located in both the southern and northern part of the study area (Figure 4). Its smoothed seasonal outbreaks peaks ranged from 26 (95% CI: [7, 56]) to 131 cases/100,000 person-weeks (95% CI: [51, 248]) (Figure 5, Table 3).

The low-incidence pattern (low pattern) consisted of a set of 466 villages with the lowest average incidence over the study period (3 cases/1000 person-years), mainly located in the northern part of the study area (Figure 4). Its smoothed seasonal outbreaks peaks ranged from 7 (95% CI: [2, 16]) to 34 cases/100,000 person-weeks (95% CI: [7, 81]) (Figure 5, Table 3).

The two higher-incidence patterns (high and intermediate) correspond to 23% of the population and 19% of the villages.

The observed incidence of the patterns, their smoothed incidence and their 95% point-wise confidence intervals of smoothing were highlighted for each malaria incidence pattern (Figure 6). In all patterns, the observed incidence rates were within the ranges except for a few peaks in the high pattern (Figure 6).

### 3.3. Velocity and Acceleration of Malaria Incidence Patterns

The velocities and accelerations (Figure A2) of the high pattern were higher, followed by those of the intermediate pattern, and those of the low pattern were the lowest (Figure A2). In both the growth and decline phase of malaria incidence patterns, when velocity and acceleration functions had the same sign in an interval, then malaria incidence patterns were in an acceleration situation; and when they had opposite signs, malaria incidence patterns, they were in a slowdown (deceleration) situation. For example, between the dates of the onset (A) and the slowdown (C) of seasonal outbreaks of patterns, velocity and acceleration functions were both positives, so malaria incidence patterns were increasing rapidly. Between the dates of slowdown (C) and the peak (D), velocity functions were positives while acceleration functions were negatives, malaria incidence patterns were also increasing but slowly until achieving the peak.

Of the 3 malaria incidence patterns, there were a total of 15 seasonal outbreaks. Each pattern had five seasonal outbreaks, which corresponded to the seasonal outbreaks that started in each year of the study period (from 2008 to 2012). The dates corresponding to the seven epidemiological indicators derived from the velocity and acceleration functions (Figure 7), as described in the methodology, were determined for all seasonal epidemics starting from 2008 to 2011. For the seasonal outbreak starting in the year 2012, the dates of the indicators characterizing the beginning of the end (F) and the end of the seasonal epidemics (G) were not determined because the study period ended in December 2012 (Table 4, Figure 7).

The results (Table 4) showed that the low pattern was always the one that started (A) the latest. The high pattern started three times earlier during the five seasonal outbreaks and the intermediate pattern twice earlier. In addition, seasonal outbreaks of the high and intermediate patterns usually started between April and June with a lag between 1 and 3 weeks. Those of the low pattern started between June and July with a delay between 4 and 9 weeks after the intermediate pattern, and with a lag between 3 and 10 weeks after the high pattern (Table 4).

The phases of pre-slowdown (B) and slowdown (C) of epidemic’s growth started mainly between August and September for all patterns with a lag between 1 and 2 weeks. Then, the peak (D) of seasonal outbreaks for all patterns, occurred between October and November almost at the same time or with a maximum of 1 week lag. The beginning of the deceleration phase of the decrease (E) occurred between November and December for all patterns, almost at the same time or with maximum 1 week of lag. The exception to the latter point was the E of seasonal outbreaks beginning in 2009 and 2010 of the high pattern, and those beginning in 2010 of the intermediate pattern began between January and February of their following years, respectively (Table 4).

The tails (F) of seasonal outbreaks for low pattern were the earliest, starting in December; those of high pattern were the latest, starting between December and March. Those of the intermediate pattern followed those of the high pattern and started between December and February. Moreover, the lag between high and low pattern was from 1 to 11 weeks, those between high and intermediate pattern was from 1 to 9 weeks and those between intermediate and low pattern was from 0 to 7 weeks (Table 4).

The end of seasonal outbreaks (G) for the high and intermediate pattern occurred between March and May with a lag from 0 to 7 weeks; those of low pattern occurred the earliest between February and March with a lag from 3 to 13 weeks before high pattern and a lag from 3 to 9 weeks before the intermediate pattern.

The seasonal outbreaks for all patterns were further described with the PCA performed on the durations between selected relevant epidemiological indicators (Figure 8, Panel A). These were the duration of strict growth’s acceleration phase (AB); those between start and peak (AD); those between slowdown of growth and decline (CE) indicating the width of the peak area; those between peak and the end (DG); those between the tail and the end of seasonal outbreaks (FG); those between pre-slowdown and the tail (BF) indicating the intermediate width of epidemics; those between the start and the end of epidemic episodes (AG) indicating the duration of the seasonal outbreaks. 

The result of PCA (Figure 8, Table A2) showed that the seasonal outbreaks (Figure 8, Panel B) of high pattern starting since 2009 (2009H) and 2010 (2010H) and those of the intermediate pattern starting since 2010 (2010I) were mainly characterized by a high BF and CE, and also by a low FG. 

In addition, the seasonal outbreaks of low pattern were characterized by low AG, DG, AD, and AB. The seasonal outbreaks starting since 2008 and 2011 for high and intermediate patterns (2008H, 2008I, 2011I, and 2011H) were mainly characterized on the one hand by high FG and on the other hand by low BF and CE. In addition, 2008H, 2009I, and 2011H were also characterized by a high AG, AD, AB, and DG.

## 4. Discussion

The approach used here led to the identification of three distinct patterns for the time-course of malaria incidence in a village, by taking into account dynamics of malaria incidence over the whole study period. In addition, this work allowed the determination of epidemiological indicators based on the velocities and accelerations of these incidence patterns, characterizing the seasonal outbreaks of the patterns.

The choice of dissimilarity measure for functional data is important before applying an unsupervised classification method, to have well-separated classes. Some other dissimilarity measures could be added [29]. We preferred to limit them on the measures less dependent to the autocorrelation structure. Indeed, the smoothing approach of functional data may impact the autocorrelation structure. For the choice of validity indices, we preferred also to concentrate on a small number of those assessing the separability (Dunn), compactness (connectivity), the quality of clustering for villages in average (silhouette) [34,35], and percentage of inertia (R^2^). 

The detection methods of transmission foci or hotspots have been defined differently in the literature [37]. There are methods that define them from an incidence or prevalence threshold [15], others with biological parameter [38], and others from scanning algorithm [10] or geostatistical approaches [9]. In addition, spatial and temporal analyses were often based on the fragmentation of the study period. Indeed in some researches, these temporal divisions were based on the calendar (month, year) or the rainy seasons, in other works of temporal fragmentation, methods were based on algorithms such as change point analysis [15,16,17,18,19,39,40]. 

In our study, patterns identification was made by taking into account not only the value of the incidence but also the dynamic of the malaria incidence over the whole study period, hence the malaria incidence pattern term. Consequently, this method can be used to distinguish two spatial units that have the same level of incidence or the same number of cases, but with different dynamics. Indeed, an epidemic that starts with a high intensity and declines over time is different from another that increases over time, leading to different control strategies.

With our approach, characterizations of the seasonal outbreaks have been made using the velocities and accelerations of the malaria incidence pattern. This allowed us to define epidemiological indicators for which seasonal outbreaks were further described. The results showed that the low-incidence pattern was the latest to start and the earliest to end seasonal outbreaks, all incidence patterns reached their seasonal peaks almost at the same time. In the case of other countries, different results can be found with these epidemiological indicators where, for example, seasonal peaks would be reached at significantly different times. 

Furthermore, malaria control strategies are usually implemented at the beginning or middle of the rainy season [23,41,42,43,44]. In Senegal, the beginning of the rainy season is generally between May and June. However, our results showed that seasonal epidemics could start from April in the high and intermediate patterns. All these particularities could guide political actors on the priority to be given to the first dates and places of intervention to cushion the impact that the epidemic could have. In addition, knowledge of the other indicators and their durations, such as the peak area, could guide the refinement of strategies according to the characteristics of the patterns for a rapid decline and end of the epidemic.

Moreover, the seasonal outbreaks 2009H, 2010H, and 2010I were remarkable. These seasonal epidemics were mainly characterized by a large peak area (CE) and a large intermediate width of the epidemic (BF) but also by a short end of epidemic phase (FG). An in-depth analysis of their velocities and accelerations showed that the acceleration of phase decline (DE) was not direct on these seasonal epidemics, since they were disrupted by a small slowdown phase indoors. Indeed, during the DE phases of these three particular seasonal epidemics, there was exceptionally a moment when the velocity was negative and the acceleration positive (which translates into a slowdown), then the acceleration became negative again (still while the velocity was negative) to continue its phase of acceleration of the decline. This would potentially partly explain these large widths. Despite this, their ends of epidemic phases were short, on the one hand, by a late onset of the beginning of the end of the epidemic (F).

Furthermore, researches had focused on the search for epidemic thresholds and stratification into intensity levels of different epidemics, particularly in the field of influenza surveillance and acute respiratory infection in Europe [45,46,47,48,49]. However, as stated by numerous authors, there was no automatic and objective way to compare thresholds and intensity levels across the studied countries. Although the epidemiological contexts are not the same with malaria, we were able to introduce an approach based on functional data allowing the smoothing of the time series of the village incidence by a single smoothing parameter allowing a possible comparison between them since they had the same scale [4]. For this purpose, even if this was not our main objective, we could define the starting date of an outbreak as the time from which the velocity and acceleration functions are strictly positive for at least three consecutive weeks (indeed, the first symptoms of malaria appear 1 to 4 weeks after infection [50]). Thus, this approach can be applied in other disease contexts.

Moreover, this work had shown that villages belonging to the same pattern are not necessarily grouped geographically. This is not very surprising given that the identification of the patterns was based solely on their temporal dynamics. Thus, a relatively small number of high-incidence villages were adjacent to low-incidence villages. It may be useful to investigate social and environmental factors that may be associated with locally high incidence (e.g., proximity to water bodies, use of control measures, etc.). The two higher-incidence patterns correspond to 23% of the population. Awareness of these trends may assist district health teams to strengthen control in high-risk communities and guide targeted intervention, and our results suggest that a targeted strategy may need to include about 20% of the population.

## 5. Conclusions

The approach used here led to the identification of three distinct malaria patterns in west-central Senegal, by considering their temporal dynamics. Epidemiological indicators derived from the velocities and accelerations of these patterns, may be useful to guide targeted control measures according to the characteristics of the patterns. 

## Figures and Tables

**Figure 1 ijerph-17-04168-f001:**
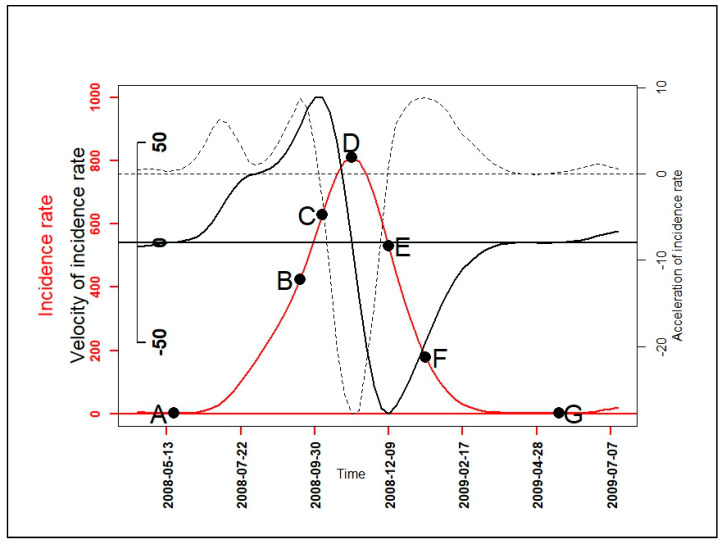
A graphical example for the seven epidemiological indicators: the beginning of seasonal outbreaks and the start acceleration of the growth phase (A); the beginning of the pre-slowdown of the growth phase (B); the deceleration’s beginning of growth phase (C); the peak (D) also corresponding after to the beginning of the acceleration of the decrease phase; the beginning of the deceleration of the decrease phase (E); the beginning of the tail (F); the end of the seasonal outbreaks (G); functional incidence in red line, functional velocity in black bold line (first derivative), and functional acceleration in black discontinuous line (second derivative).

**Figure 2 ijerph-17-04168-f002:**
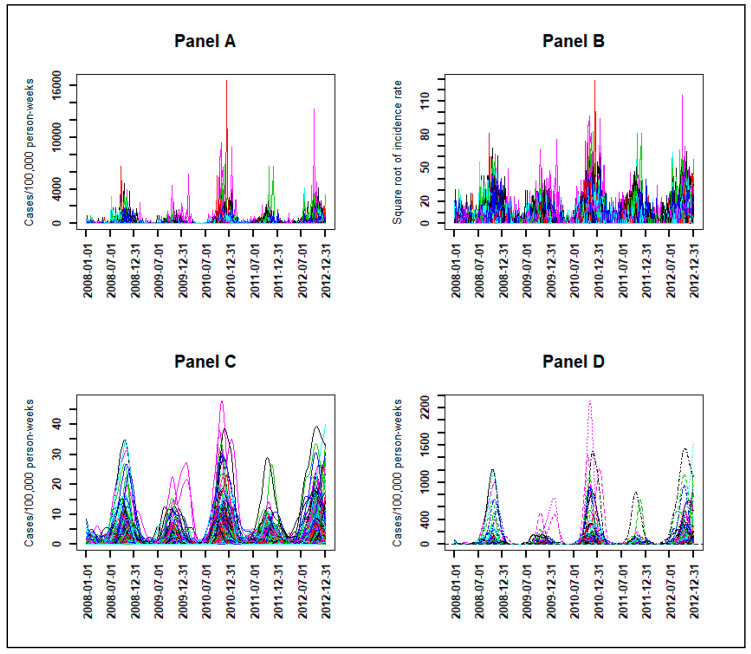
Weekly evolution of malaria incidence for each village from January 2008 to December 2012: observed time series (**Panel A**) and at the square root scale (**Panel B**), smoothed time series at the square root scale (**Panel C**) and at the scale of untransformed observations (**Panel D**).

**Figure 3 ijerph-17-04168-f003:**
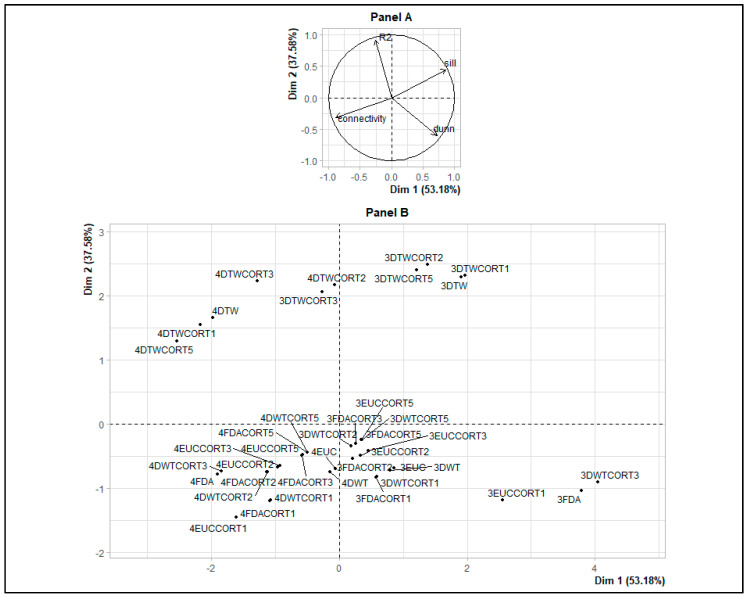
Principal component analysis on validity indices and dissimilarity measures for 3 and 4 number patterns: validity indices map (Variables, **Panel A**), dissimilarity measures map (Individuals, **Panel B**). 4DTWCORT3 is the assessed hierarchical ascending clustering (HAC) result with the potential number of patterns chosen as 4, and performed with DTWCORT3 dissimilarity measure ($d_{\mathrm{DTWCORT}3}$) taking into account the 9.4% of $d_{DTW}$ (data value) and 90.5% CORT (data behavior) (Table 1, ξ = 3), 3FDA is the assessed HAC result with the potential number of patterns chosen as 3, and performed with $d_{FDA}$ dissimilarity measure taking into account 100% of data value, etc.

**Figure 4 ijerph-17-04168-f004:**
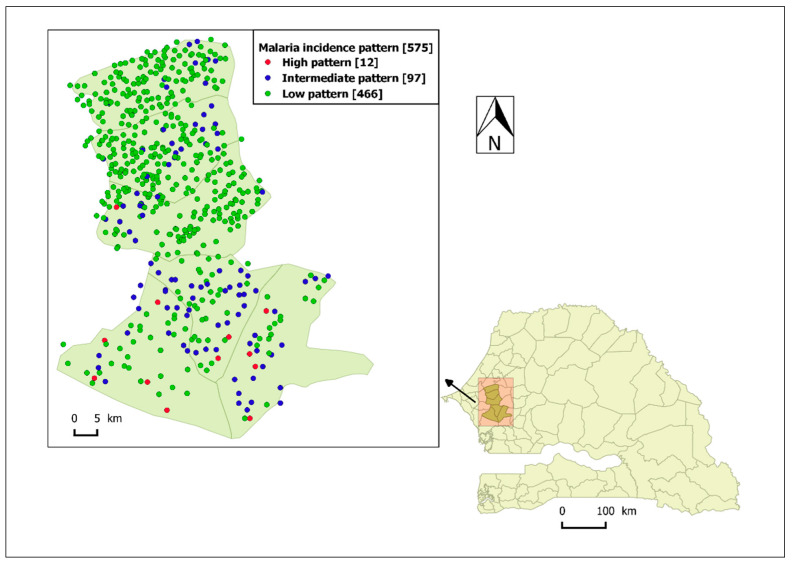
The spatial distribution of malaria incidence pattern villages in the study area: Senegal map and the location of the study area pointed by the arrow, high-incidence pattern villages in red dot, intermediate-incidence pattern villages in blue dot, and low-incidence pattern villages in green dot.

**Figure 5 ijerph-17-04168-f005:**
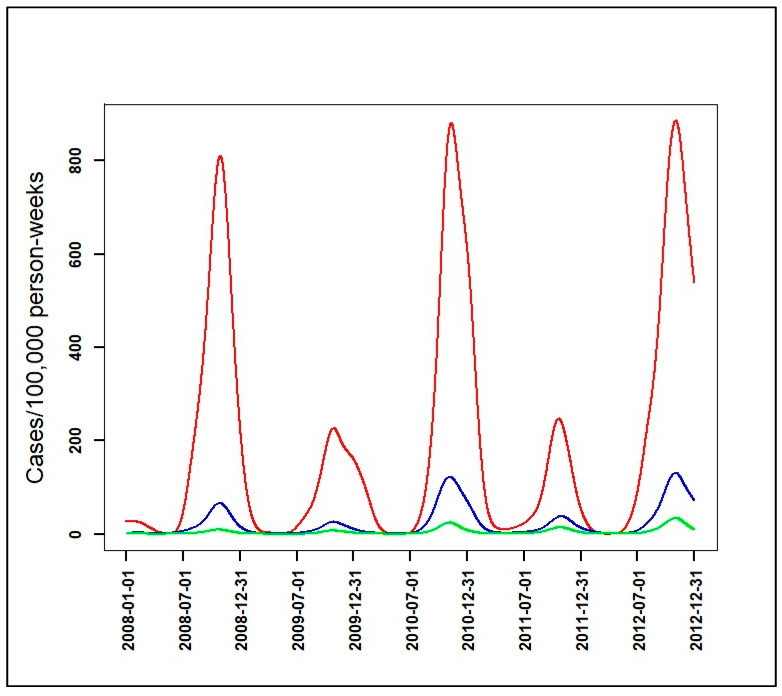
The smoothed functions (functional data) for each malaria incidence pattern between January 2008 to December 2012: high-incidence pattern in red line, intermediate-incidence pattern in blue line, and low-incidence pattern in green line.

**Figure 6 ijerph-17-04168-f006:**
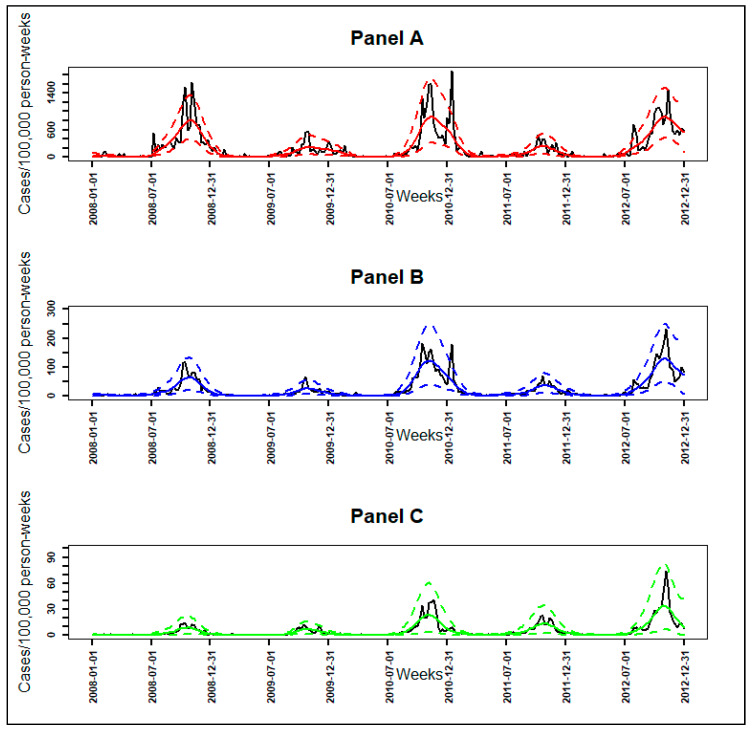
Weekly observed malaria incidence in black solid line, smoothed malaria incidence in color solid line, and smooth 95% point-wise confidence intervals in discontinuous color line: high-incidence pattern in red (**Panel A**), intermediate-incidence pattern in blue (**Panel B**), and low-incidence pattern in green (**Panel C**).

**Figure 7 ijerph-17-04168-f007:**
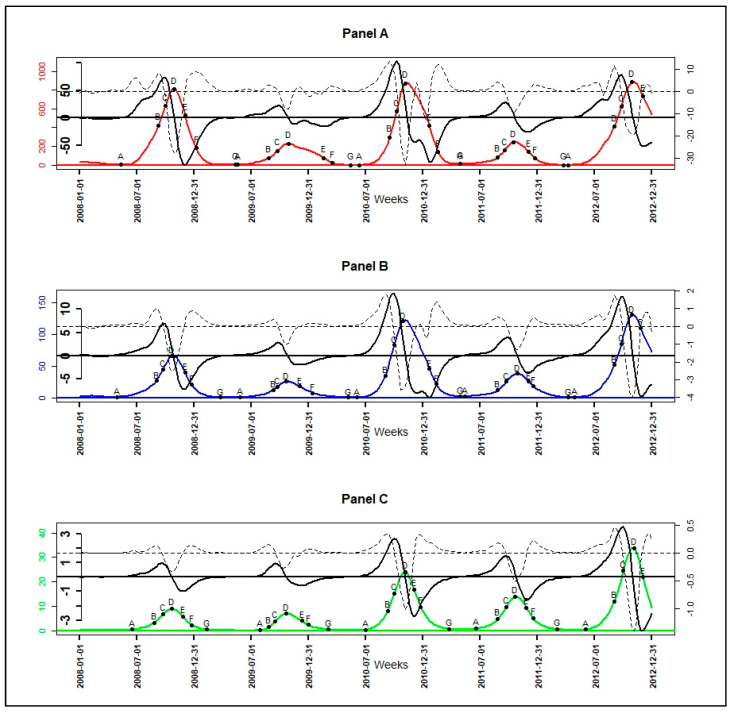
Smoothed incidence in color solid line, their velocity in black bold solid line, their acceleration in black discontinuous line, and the epidemiological indicator of their seasonal outbreaks (A: onset, B: near slowdown of growth, C: beginning slowdown of growth, D: peak, E: beginning acceleration of decline, F: beginning of tail, G: end): high-incidence pattern in red (**Panel A**), intermediate-incidence pattern in blue (**Panel B**), and low-incidence pattern in green (**Panel C**).

**Figure 8 ijerph-17-04168-f008:**
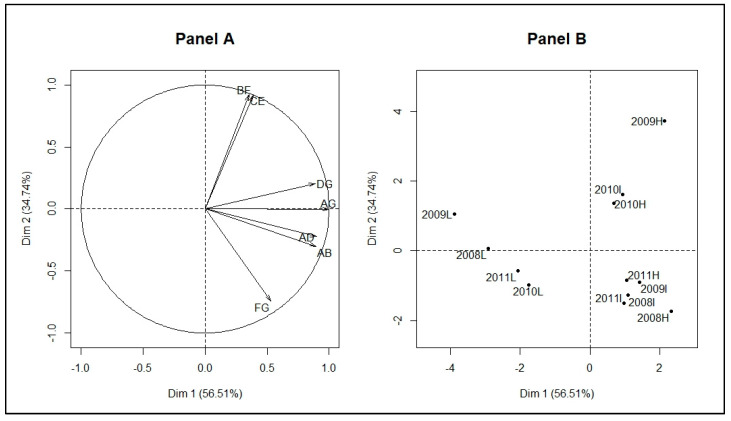
Principal component analysis on duration epidemiological indicators and seasonal outbreaks of the patterns: epidemiological indicator map (Variables, **Panel A**) (the duration of strict growth’s acceleration phase (AB); the duration between start and peak (AD); the duration between slowdown of growth and decline (CE) indicating the width of the peak area; the duration between peak and the end (DG); the duration between the tail and the end of seasonal outbreaks (FG); the duration between pre-slowdown and the tail (BF) indicating the intermediate width of epidemics; the duration between the start and the end of epidemic episodes (AG) indicating the duration of the seasonal outbreaks); seasonal outbreaks of the patterns map (Individuals, **Panel B**) (L=Low, I=Intermediate, H=High, 2009L is the seasonal outbreak starting in year 2009 in the malaria low-incidence pattern).

**Table 1 ijerph-17-04168-t001:** The percentage of contribution in $d_{CORT}$ dissimilarity measure according to the parameter ξ.

ξ	Behavior Contribution (%)	Values Contribution (%)
0	0	100
1	46.2	53.7
2	76.2	23.8
3	90.5	9.4
≥5	~100	~0

ξ is a non-negative parameter of the adaptative function $f_{\xi }\left(u\right)=\frac{2}{1+\exp\left(\xi u\right)}$ in the $d_{CORT}$ dissimilarity measure (5).

**Table 2 ijerph-17-04168-t002:** The description of epidemiological indicators and the determination of their corresponding date for a functional data $C_{q}$ of pattern *q*.

Epidemiological Indicators (EI)	Determination of EI’s Dates
Beginning of seasonal outbreaks and the start acceleration of the growth phase (A)	$t_{A}=\{\begin{array}{c} first\;t\;such\;C_{q}{}^{\prime }\left(t\right)>0\\ \;\;C_{q}{}^{\prime \prime }\left(t\right)>0\\ on\;3\;weeks \end{array}$
Beginning of the pre-slowdown of the growth phase (B)	$t_{B}=\{\begin{array}{c} \;\underset{t\;such\;C_{q}{}^{\prime }\left(t\right)>0}{\mathrm{argmax}}\left(C_{q}{}^{\prime \prime }\left(t\right)\right) \end{array}$
Deceleration’s beginning of growth phase (C)	$t_{C}=\{\begin{array}{c} \;\underset{t\;such\;C_{q}{}^{\prime \prime }\left(t\right)=0}{\mathrm{argmax}}\left(C_{q}{}^{\prime }\left(t\right)\right) \end{array}$
Peak of seasonal outbreaks and beginning of the acceleration of the decrease phase (D)	$t_{D}=\{\begin{array}{c} C_{q}{}^{\prime }\left(t\right)=0\;\;\;\;\;\;\;\;\;\;\;\;\;\;\\ C_{q}{}^{\prime \prime }\left(t\right)<0\;\;\;\;\;\;\;\;\;\;\;\;\; \end{array}$
Beginning of the deceleration of the decrease phase (E)	$t_{E}=\{\begin{array}{c} \;\;\underset{t\;such\;C_{q}{}^{\prime \prime }\left(t\right)=0}{\mathrm{argmin}}\left(C_{q}{}^{\prime }\left(t\right)\right) \end{array}$
Beginning of the tail of seasonal outbreaks (F)	$t_{F}=\{\begin{array}{c} \;\underset{t\;such\;C_{q}{}^{\prime }\left(t\right)<0}{\mathrm{argmax}}\left(C_{q}{}^{\prime \prime }\left(t\right)\right) \end{array}$
End of seasonal outbreaks (G)	$t_{G}=\{\begin{array}{c} first\;t\;such\;C_{q}{}^{\prime }\left(t\right)=0\;\\ on\;3\;weeks \end{array}$

**Table 3 ijerph-17-04168-t003:** Incidence description of malaria incidence patterns: the type of pattern, their number of villages, and their ranges peaks of smoothed seasonal outbreaks with 95% CI and their observed cumulative incidence over the five years of the study period.

Malaria Incidence Patterns	Number of Villages	Range Peaks of Smoothed Seasonal Outbreaks (Cases/100,000 Person-Weeks) with [95% CI]	Observed Cumulative Incidence over the Five Year-Study Period (Cases/1000 Person-Years)
High	12	227 [65, 487]–884 [420, 1518]	114
Intermediate	97	26 [7, 56]–131 [51, 248]	13
Low	466	7 [2, 16]–34 [7, 81]	3

**Table 4 ijerph-17-04168-t004:** The epidemiological indicators (EI) and their characteristics over seasonal outbreaks.

Start Year Seasonal Outbreak	EI	DateHigh	DateInter	DateLow	WeekHigh	WeekInter	WeekLow
2008	A	13/05/2008	29/04/2008	17/06/2008	20	18	25
2009	A	19/05/2009	26/05/2009	28/07/2009	21	22	31
2010	A	08/06/2010	01/06/2010	29/06/2010	24	23	27
2011	A	26/04/2011	10/05/2011	14/06/2011	18	20	25
2012	A	03/04/2012	24/04/2012	29/05/2012	15	18	23
2008	B	09/09/2008	02/09/2008	26/08/2008	37	36	35
2009	B	25/08/2009	08/09/2009	25/08/2009	35	37	35
2010	B	14/09/2010	31/08/2010	07/09/2010	38	36	37
2011	B	23/08/2011	23/08/2011	23/08/2011	35	35	35
2012	B	28/08/2012	28/08/2012	28/08/2012	36	36	36
2008	C	30/09/2008	23/09/2008	23/09/2008	40	39	39
2009	C	22/09/2009	22/09/2009	15/09/2009	39	39	38
2010	C	05/10/2010	28/09/2010	28/09/2010	41	40	40
2011	C	13/09/2011	20/09/2011	20/09/2011	38	39	39
2012	C	18/09/2012	18/09/2012	25/09/2012	39	39	40
2008	D	28/10/2008	21/10/2008	21/10/2008	44	43	43
2009	D	27/10/2009	20/10/2009	20/10/2009	44	43	43
2010	D	02/11/2010	26/10/2010	02/11/2010	45	44	45
2011	D	11/10/2011	25/10/2011	18/10/2011	42	44	43
2012	D	23/10/2012	23/10/2012	30/10/2012	44	44	45
2008	E	02/12/2008	02/12/2008	25/11/2008	49	49	48
2009	E	16/02/2010	01/12/2009	08/12/2009	8	49	50
2010	E	18/01/2011	18/01/2011	30/11/2010	4	4	49
2011	E	29/11/2011	29/11/2011	22/11/2011	49	49	48
2012	E	27/11/2012	20/11/2012	27/11/2012	49	48	49
2008	F	06/01/2009	23/12/2008	23/12/2008	2	52	52
2009	F	16/03/2010	12/01/2010	29/12/2009	12	3	1
2010	F	15/02/2011	08/02/2011	21/12/2010	8	7	52
2011	F	20/12/2011	13/12/2011	13/12/2011	52	51	51
2008	G	12/05/2009	24/03/2009	10/02/2009	20	13	7
2009	G	11/05/2010	04/05/2010	02/03/2010	20	19	10
2010	G	26/04/2011	26/04/2011	22/03/2011	18	18	13
2011	G	20/03/2012	03/04/2012	28/02/2012	13	15	10

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
