# Peer review of "Application of Functional Data Analysis to Identify Patterns of Malaria Incidence, to Guide Targeted Control Strategies"

_ijerph, 2020, doi:10.3390/ijerph17114168_

Round 1
Reviewer 1 Report
This manuscript presents results from a study that employs a functional data analytic approach for identifying patterns of malaria incidence from 575 villages in west-central Senegal. The authors approximate malaria incidence curves from these villages (from January 2008 – December 2012) using a B-spline basis representation and then consider a number of dissimilarity metrics based on the values of the approximated curves and other properties (including rate of change, acceleration, peaks, etc.) to perform a hierarchical clustering of the curves. They identified 3 clusters of curves corresponding to high, intermediate, and low incidences over time. While the functional data analytic approach seems warranted, the authors’ specific choices for carrying out this approach could be better explained/justified in the manuscript. Key details and rationale are omitted and it is unclear how this functional data analytic approach is superior to more classic time-series or other multivariate approaches that could have been utilized. Specific comments/questions are given below.
- In equation (2), it is more appropriate to say that x_i(t) is approximated by the finite sum.
- Equation after line 93 has two equation labels (3) and (4) – remove (4).
- Line 94: lambda is a single smoothing parameter.It is often explicitly stated that this value is non-negative.
- Equation (4) – what is “P” under the integral sign?It is not defined.
- It would be helpful to briefly explain the role of the smoothing parameter and penalty to the reader who is likely new to functional data analysis.Something like: “The penalty term controls the smoothness of the estimate for x_(t). Large values of lambda yield nearly linear curve estimates while small values of lambda yield wiggly curve estimates.” Or something like this.
- Line 98: The authors state that the basis functions are “mathematically independent functions.”It is not clear what this means. I think the authors may be trying to say that the basis functions from a family are orthogonal, but this is not the case for all families. For example, B-spline basis functions are not orthogonal, but are commonly used in functional data analysis.
- Line 122: The authors claim that they considered 20 dissimilarity measures, what are they?At least provide the values of xi that correspond.
- Line 126: Authors say that they are searching for the number of patterns with the “best” dissimilarity measure – I assume “best” means smallest – but it is not clear.
- In lines 124 – 130, the authors mention several methods for performing and evaluating the functional clustering.It might be helpful to give the reader a short summary of each method that is mentioned.
- Lines 129 – 130: It is unclear how a principal components analysis is being used on the indices to select the best classification.
- Line 148: Again, it would be helpful to explain why PCA is used here.
- Overall, section 2.2 is rather hard to follow.Readers would benefit from more detail (does not need to be extensive) on all of the different methods and indices that are mentioned.
- Based on Figure 2, it seems that using a wavelet basis rather than a B-spline basis would be more appropriate since the curves have highly localized features – which is ideal for a wavelet-based representation.Can the authors comment on why they chose to use B-spline basis functions rather than wavelet basis functions to represent the individual time series?
- In Figure 2, it seems that the functions in panel (A) are being poorly represented by the basis function approximations in panel (B) since the y-axis scale in panel (B) has a maximum value of 2400 whereas panel (A) has a maximum value of 17000.Can the authors comment on this?
- In hierarchical clustering, structures having clusters with fewer than 5% of the sample in them are often discarded for a structure with a lower number of clusters.Can the authors comment on why they chose to keep the 3-cluster model vs. a 2-cluster model?
Author Response
Point 1: This manuscript presents results from a study that employs a functional data analytic approach for identifying patterns of malaria incidence from 575 villages in west-central Senegal. The authors approximate malaria incidence curves from these villages (from January 2008 – December 2012) using a B-spline basis representation and then consider a number of dissimilarity metrics based on the values of the approximated curves and other properties (including rate of change, acceleration, peaks, etc.) to perform a hierarchical clustering of the curves. They identified 3 clusters of curves corresponding to high, intermediate, and low incidences over time. While the functional data analytic approach seems warranted, the authors’ specific choices for carrying out this approach could be better explained/justified in the manuscript. Key details and rationale are omitted, and it is unclear how this functional data analytic approach is superior to more classic time-series or other multivariate approaches that could have been utilized. Specific comments/questions are given below.
Response 1: First, thank you for reviewing our manuscript.
We apologize for any lack of clarity in the reasons related to specific choices. The rate of change (velocity), acceleration, peaks were determined after the smoothing and clustering steps.
We have better clarified the reasons of our choice in introduction section line 37-47 as :
“The development of technology has increasingly enabled the use of sophisticated tools to collect and store large amounts of complex data, particularly in scientific fields. These data are often continuous but observed over a finite number of points (discretization points) [1–3]. This is the case for meteorological data, electrocardiogram, time series, growth curves, for example. Functional data approach would be better adapted to handle these data by taking into account some of their particularities. Indeed, this approach is useful to handle a large sample of spatial unit (village) allowing comparison between them and to reduce data dimension (number of observations) for long time series. In addition, the number of observations may be higher than the size of sample making difficult to analyse. The observations are not always observed at regular time lag (every hour, every day etc.) and this latter may differ from one place to another [1, 3]. Moreover, the use of functional data allows also estimating the velocity and acceleration of time series.”
We provide below point-by-point responses to all specific questions and comments.
Point 2: In equation (2), it is more appropriate to say that x_i(t) is approximated by the finite sum.
Response 2: Thank you for this suggestion. We have used it and added precision in line 136 as “The function x_i is approximated by a finite sum of linear combination of basis functions”.
Point 3: Equation after line 93 has two equation labels (3) and (4) – remove (4).
Response 3: We have corrected it. Thank you.
Point 4: Line 94: lambda is a single smoothing parameter.It is often explicitly stated that this value is
non-negative.
Response 4: You are right. This information is added in line 141 as “where λ is the non-negative smoothing parameter (3)…”.
Point 5: Equation (4) – what is “P” under the integral sign?It is not defined.
Response 5: We are sorry for this, you are right. P is defined in line 141-142 as “P the studied period expressed here as [1, 261] in which times are continuous”.
Point 6: It would be helpful to briefly explain the role of the smoothing parameter and penalty to the
reader who is likely new to functional data analysis. Something like: “The penalty term
controls the smoothness of the estimate for x_(t). Large values of lambda yield nearly linear
curve estimates while small values of lambda yield wiggly curve estimates.” Or something
like this.
Response 6: Thank you for this suggestion. The role of the smoothing parameter and penalty is defined in line 145-147 as “The penalty term (4) controls the smoothness of the estimate for x_i (t). Large values of lambda λ yield nearly linear curve estimates while small values of lambda yield wiggly curve estimates getting closer to observed data.”
Point 7: Line 98: The authors state that the basis functions are “mathematically independent
functions.”It is not clear what this means. I think the authors may be trying to say that the
basis functions from a family are orthogonal, but this is not the case for all families. For
example, B-spline basis functions are not orthogonal, but are commonly used in functional
data analysis.
Response 7: To simplify we have removed “mathematically independent” which has not any consequence on the definition and usefulness of basis functions.
Point 8: Line 122: The authors claim that they considered 20 dissimilarity measures, what are they?
At least provide the values of xi that correspond.
Response 8: Thank you. We were afraid we present a too technical manuscript; this was why we had not deeply described some parts.
Table 1 is based on theoretical values from its original article research. We have added the reference in line 185-186 as “The Table 1 comes from the original article [30] which developed the d_CORT dissimilarity measure.”.
We have added in line 189-194 “Thus, a total of 20 dissimilarity measures were used in this analysis. For example, the d_EUCCORT1 is the dissimilarity measure based on temporal correlation (CORT) and the Euclidean distance (d_EUC) (equation 5) when ξ is equal to 1 (Table 1), this considered 46.2% of behaviour contribution (given by CORT) and 53.7% of values contribution (given by d_EUC). The others dissimilarity measures were obtained in the same way and when ξ=0 (Table 1) we had the 4 dissimilarity measures based only on values.”
Point 9: Line 126: Authors say that they are searching for the number of patterns with the “best”
dissimilarity measure – I assume “best” means smallest – but it is not clear.
Response 9: Sorry for this misunderstanding. This meant the most ability dissimilarity measures to assess difference between curves. We assumed that we could use a lot of dissimilarity measures to perform hierarchical clustering, but we don’t know particularly the most able to determine difference between curves for getting a good classification.
To be clearer we have updated line 208-211 as “Thus, hierarchical ascending clustering (HAC) was performed on smooth functions using each dissimilarity measure with Ward aggregation method [32]. This was to find the most ability dissimilarity measure to assess difference between curves for reaching a better quality of classification.”
Point 10: In lines 124 – 130, the authors mention several methods for performing and evaluating the
functional clustering.It might be helpful to give the reader a short summary of each method
that is mentioned.
Response 10: This following paragraph has been added in line 215-223: “The connectivity indicates the degree of connectedness of the clusters, as determined by the k-nearest neighbors (in this work k=10). The connectivity has a value between 0 and infinity and should be minimized. Both the Silhouette Width and the Dunn Index combine measures of compactness and separation of the clusters. The Silhouette Width is the average of each curve's Silhouette value. The Silhouette value measures the degree of confidence in a particular clustering assignment and lies in the interval [-1,1], with well-clustered curves having values near 1 and poorly clustered curves having values near -1. The Dunn Index is the ratio between the smallest distance between curves not in the same cluster to the largest intra-cluster distance. It has a value between 0 and infinity and should be maximized.”
Point 11: Lines 129 – 130: It is unclear how a principal components analysis is being used on the
indices to select the best classification.
Response 11:
Hierarchical clustering (HAC)were performed using each of the 20 dissimilarity measures discussed in response 8 for the reason discussed in response 9. Then, to assess the HAC results according to potential numbers of pattern (chosen after examining of dendrograms), four validity indices (response 10) were used. To choose the final or optimal number of malaria incidence patterns, we have performed PCA on assessed HAC results to look for the one which showed the best criteria of validity indices.
We have updated line 223-227 as “To choose the final or optimal number of malaria incidence patterns, we have performed a Principal Component Analysis (PCA) [36] on assessed HAC results to look for the one which showed the best criteria of validity indices i.e. with connectivity index close to 0, high Dunn index, Silhouette Width and R2 close to 1.”
Point 12: Line 148: Again, it would be helpful to explain why PCA is used here.
Response 12: As explained in response 11, PCA were performed to look for durations which characterized seasonal outbreak of patterns.
We have updated line 246-248 as “Finally, a PCA was performed on the durations: AB, AD, CE, DG, FG, BF and AG, to look for those which characterized seasonal outbreaks of patterns.”
Point 13: Overall, section 2.2 is rather hard to follow.Readers would benefit from more detail (does
not need to be extensive) on all of the different methods and indices that are mentioned.
Response 13: We were afraid to present a too technical manuscript by explaining in details all methods and indices already described in corresponding references. We hope that the modifications proposed above clarify the manuscript.
Moreover, to make easy the understanding of our methodology, we have added at the beginning of section 2.2 a summary of the stages in line 117-126 as
“Our approach consisted of three stages.
In the first stage, we estimated malaria incidence curves (smooth functions or functional data) from these 575 villages (from January 2008 – December 2012) using a basis functions representation (village-level temporal trends).
In the second stage, hierarchical clustering was applied to the malaria incidence curves for classifying villages with similar temporal trends together. To obtain an optimal classification, several dissimilarity measures and validity indices were used.
In the third stage, resulting patterns were characterized using pre-defined epidemiological indicators from their velocity and acceleration functions to describe the overall features of temporal patterns identified.”
Point 14: Based on Figure 2, it seems that using a wavelet basis rather than a B-spline basis would
be more appropriate since the curves have highly localized features – which is ideal for a
wavelet-based representation. Can the authors comment on why they chose to use B-spline
basis functions rather than wavelet basis functions to represent the individual time series?
Response 14: Thank you for this question. We are agree, wavelet basis could be used at the place of B-spline, but we have not tested wavelet basis representation because of the lack of R program to apply it. However, the B-spline basis functions are commonly used in cases where Fourier or Wavelet basis functions are not used for periodic data.
Point 15: In Figure 2, it seems that the functions in panel (A) are being poorly represented by the
basis function approximations in panel (B) since the y-axis scale in panel (B) has a
maximum value of 2400 whereas panel (A) has a maximum value of 17000.Can the authors
comment on this?
Response 15:
We have replaced this figure2 by the one where every details of smoothing were applied in 4 panels. Before approximating the observed data (Panel A), we have transformed them with square root function to stabilize the variability (see equation 1) (Panel B), then we had applied functional data method to estimate the curves on these transformed data (Panel C). For epidemiological interpretation, we decide to apply a square function (reciprocal function) on these functional data to have the corresponding functional data in the same scale of the observed data (Panel D).
Moreover, there are a few weeks that attempted high values of incidence in Panel A, in addition the smoothing has erased properties for some exceptional high values and the curves were smoothing with a single parameter lambda with the same number of basis functions. As a result, we are in the situation of a balance between having well smoothed curves on the same scale to compare them or handle them together, and the individual excellent smoothing of curves with different scale and no comparison between them.
We have updated the Results sub section 3.1. From observed to smoothed malaria incidence line 265-280 as “The observed time series of malaria incidence for each of the 575 villages from January 2008 to December 2012 were determined (Figure 2, Panel A). The observed malaria incidence ranged from 0 to 183 cases/1000 person-years at the village level, and the median was 4 cases/1000 person-years with interquartile range (2, 9). At village and week levels the observed malaria incidence ranged from 0 to 17,000 cases/100,000 person-weeks (Figure 2, Panel A).
Because of the high variability, the transformation with square root function was applied to the time series of (Figure 2, Panel A) to obtain the observed time series of malaria incidence in square root scale (Panel B) as explained in method section equation (1).
With the transformed observed time series (Panel B), the search for the optimal number of basis function and the optimal smoothing parameter gave K_opt=110 and λ_opt=103 which minimized the error by GCV equal to 11.8 with a standard deviation of σ=0.12. Using these optimal parameters, the smoothed transformed time series of malaria incidence for each village were determined (Figure 2, Panel C).
For epidemiological interpretation, we applied the square function (reciprocal function) to obtain the smoothed time series of malaria incidence corresponding to the observed time series (Figure 2, Panel D).”
Note that the clustering was performed on smoothed times series of transformed data (Figure 2, Panel C). This is mentioned in line 293-294 as “Three patterns with the DTWCORT1 dissimilarity measure (3DTWCORT1 HAC result) were obtained with the application of HAC on the smoothed transformed times series (Figure 2, Panel C).”
Point 16: In hierarchical clustering, structures having clusters with fewer than 5% of the sample in
them are often discarded for a structure with a lower number of clusters. Can the authors
comment on why they chose to keep the 3-cluster model vs. a 2-cluster model?
Response 16: Thank you for this question. We agree, generally it is preferable to avoid small size sample of clusters. However, even if we chose 2 clusters (patterns), the high pattern which had the lower numbers of villages would not be grouped with another pattern, and the dendrogram showed this.
In addition, in the context of malaria heterogeneity particularly in low transmission case, small number of spatial units are responsible of the most part of transmission. So, having small numbers of spatial units in a cluster has a sense in the context of our study. Epidemiological realities and statistical tools are combined to decide in fine what to choose which has sense in epidemiology context and not in contradiction with statistic rules.
Additional comments by authors in the manuscripts:
We have added an interpretation of deceleration and acceleration situation through the velocity and acceleration function line 360-368 as “In both growth and decline phase of malaria incidence patterns, when velocity and acceleration functions had the same sign in an interval then malaria incidence patterns was in acceleration situation and when they had opposite signs malaria incidence patterns was in slowdown (deceleration) situation . For example, between the dates of the onset (A) and the slowdown (C) of seasonal outbreaks of patterns, velocity and acceleration functions were both positive, so malaria incidence patterns were increasing rapidly. Between the dates of slowdown (C) and the peak (D), velocity function was positive while acceleration function was negative, malaria incidence patterns were also increasing but slowly until achieving the peak.”
We have modified line 476-482 by "An in-depth analysis of their velocities and accelerations showed that the acceleration of phase decline (DE) was not direct on these seasonal epidemics, since they were disrupted by a small slowdown phase indoors. Indeed, during the DE phases of these 3 particular seasonal epidemics, there was exceptionally a moment when the velocity was negative and the acceleration positive (which translates into a slowdown), then the acceleration became negative again (still while the velocity was negative) to continue its phase of acceleration of the decline.”

Reviewer 2 Report
Review: Application of functional data analysis to identify patterns of malaria incidence, to guide target control strategies
In this manuscript, the authors proposed a multi-stage modelling approach for classifying temporal trends in malaria incidence, with trends estimated at the areal/village level. The approach consists of three stages. In the first stage, B-splines are fitted to observed data to estimate village-level temporal trends. In the second stage, hierarchical clustering is applied to estimated smooth functions as a way of grouping villages with similar temporal trends together. To obtain an optimal classification, several dissimilarity measures are tested. Resulting clusters are characterised in the third stage using pre-defined measures (velocity, acceleration) to describe the overall features of temporal trends identified.
The paper presents a well-motivated, creative approach to unsupervised clustering with interesting results. There were parts of the manuscript that would benefit from further clarification, particularly the second stage of modelling (hierarchical clustering of smooth functions). Suggested points for further clarification are listed below:
Lines 39-42: ‘Existing approaches used, to target malaria risk areas are based on aggregated incidence and prevalence rate, [2–8] in discrete time periods [9–13].In contrast, the functional data approaches the whole time series of each spatial unit, giving potentially a better understanding of the underlying temporal patterns’. The authors imply that a point of difference motivating the application of functional data approaches is that data need not be aggregated into discrete time periods. However, subsequent methodology presented in Section 2.2.1 uses data aggregated by week.
Line 54: ‘the velocity and acceleration’. An additional sentence interpreting these terms in the context of the data being analysed would be useful here. The authors provide the example of lags between onset, peaks and end dates, but it was unclear how these features could be inferred from velocity and acceleration.
Lines 104-105. Hierarchical ascending clustering: A brief description of this method and why it was chosen would strengthen this section. For example, why was HAC chosen over the k-means algorithm?
Line 106: ‘value and behaviour’: what is meant by these two terms? For readers unfamiliar with the chosen clustering method, the methodology becomes difficult to follow from this point. Should the authors decide to add a description of the clustering method (see point above), these terms could be described at the same time. Providing this further explanation would also assist with subsequent text about the different dissimilarity measures (lines 107-110).
Line 115: Table 1: Are the numbers listed in Table 1 based on the data or are these theoretical values? Further clarification is required here, in the text and/or caption for Table 1. If these values are based on the real data, Table 1 would be better placed in the results.
Line 134: ‘Q patterns’. How was Q chosen?
Line 184: Figure 3, Panel B. A brief description of codes in this plot would be helpful here (e.g. 3FDA)
Author Response
Point 1: In this manuscript, the authors proposed a multi-stage modelling approach for classifying temporal trends in malaria incidence, with trends estimated at the areal/village level. The approach consists of three stages. In the first stage, B-splines are fitted to observed data to estimate village-level temporal trends. In the second stage, hierarchical clustering is applied to estimated smooth functions as a way of grouping villages with similar temporal trends together. To obtain an optimal classification, several dissimilarity measures are tested. Resulting clusters are characterised in the third stage using pre-defined measures (velocity, acceleration) to describe the overall features of temporal trends identified.
The paper presents a well-motivated, creative approach to unsupervised clustering with interesting results. There were parts of the manuscript that would benefit from further clarification, particularly the second stage of modelling (hierarchical clustering of smooth functions). Suggested points for further clarification are listed below.
Response 1: First, thank you for reviewing our manuscript. Suggested points are replied below.
Point 2: Lines 39-42: ‘Existing approaches used, to target malaria risk areas are based on
aggregated incidence and prevalence rate, [2–8] in discrete time periods [9–13].In contrast,
the functional data approaches the whole time series of each spatial unit, giving potentially a
better understanding of the underlying temporal patterns’. The authors imply that a point of
difference motivating the application of functional data approaches is that data need not be
aggregated into discrete time periods. However, subsequent methodology presented in
Section 2.2.1 uses data aggregated by week.
Response 2: We are sorry of this confusion. First, we updated the Introduction section by adding and moving some parts to be clearer.
Then, we would like to highlight the difference between “discrete time” and “continuous time”. The useful time precision depends on the disease. In the case of rapid outbreaks (e.g. ebola, cholera), daily data are necessary. In the case of malaria (needing a development cycle in mosquitoes, and also particular development phases in human for gametocyte production), the weekly number of cases is more appropriate. While daily and weekly data are both discrete time, continuous time approaches are useful for dynamic analyses.
In addition, the aggregation we mentioned in existing approaches of targeted malaria risk areas was based on large sub-periods as rainy seasons, semesters, years independently of the useful time precision for the disease.
We have modified this paragraph in line 57-62 as:
“Existing approaches used, to target malaria risk areas are based on aggregated incidence and prevalence rate, [9–15] in large discrete time sub-periods [16–20]. Thus, malaria risk areas were identified every rainy season or every year or another large sub-period and sometimes the status at malaria risk of areas between sub-periods can change. These approaches do not provide information about the trend or temporal dynamic of malaria and continuous time approaches are useful for dynamic analysis.”
Point 3: Line 54: ‘the velocity and acceleration’. An additional sentence interpreting these terms in
the context of the data being analysed would be useful here. The authors provide the example of lags between onset, peaks and end dates, but it was unclear how these features could be inferred from velocity and acceleration.
Response 3: The first derivative is the velocity function which gave over time information about when the malaria incidence increased (growth phase period) or decreased (decline phase period). The second derivative is the acceleration function i.e. the variation of epidemic speed (velocity). This indicated how malaria incidence increased or decreased over time: quickly or slowly. Thus, temporal variations of velocity and acceleration together gave information about the malaria dynamic.
About the question of how these features (lags between onset, peaks, end dates) could be inferred from velocity and acceleration, this is based on mathematical properties of derivatives as defined in Table 2. Example the date of the peak (D) corresponded to the moment where the first derivative or velocity function (C’(t)) cancelled and the second derivative or acceleration function (C’’(t)) was negative. After determining all dates of pattern peaks, the lags between them were the difference of dates. This was the same for the others epidemiological indicators (onset, ends etc.) defined in Table 2.
To be clearer, we have updated in line 71-79 as “In addition, for further understanding of malaria incidence dynamic, the velocity, and the acceleration (velocity variation) are useful. Indeed, the velocity is the first derivative function which gives information over time about when the malaria incidence increases (growth phase period) or decreases (decline phase period). The acceleration i.e. the variation of epidemic speed (velocity) is the second derivative function. This indicates how malaria incidence increases or decreases over time: quickly or slowly [21,22]. Thus, temporal variations of velocity and acceleration together provide information about the malaria dynamic. Moreover, key features of malaria dynamic derived from velocity and acceleration functions as onsets, peaks, ends and their lags between patterns are useful to refine targeted intervention schedule.”
Point 4: Lines 104-105. Hierarchical ascending clustering: A brief description of this method and
why it was chosen would strengthen this section. For example, why was HAC chosen over the k-means algorithm?
Response 4: Thank you. A brief description is added in line 161-167 as: “Hierarchical ascending clustering [27] is one of the most popular unsupervised clustering algorithms grouping similar elements such that the elements in the same group are more similar to each other than the elements in the other groups. At the beginning each element is a cluster, then elements are grouped according to a dissimilarity measure and aggregation criteria until having one cluster grouping all elements. The advantage with this clustering method is its ability to work without prior number of clusters which can influence the results compared to Kmeans for example. Another advantage is its dendrogram tool showing different cluster possibilities.”
Point 5: Line 106: ‘value and behaviour’: what is meant by these two terms? For readers unfamiliar
with the chosen clustering method, the methodology becomes difficult to follow from this point. Should the authors decide to add a description of the clustering method (see point above), these terms could be described at the same time. Providing this further explanation would also assist with subsequent text about the different dissimilarity measures (lines 107-110).
Response 5: These following sentences are added in line 168-194:
“To perform a hierarchical ascending clustering on smooth functions (functional data or curves), a dissimilarity measure between them was necessary to assess their proximity before each grouping step. In the context of time series, several dissimilarity measures are proposed in the literature [28, 29]. This work focused on those based only on the data value or level intensity, and those based in addition, on the temporal evolution or behaviour of data over time that would adapt to the functional data [26, 28–30].
For those based only on data values we have selected 4: the Euclidean distance [29] ( ) based on the point-to-point differences between observations of the 2 curves, the Lp-metric [26] estimating the surface between 2 functional data (curves) ( ), the dynamic time warping [31] ( ) providing a measure of distance insensitive to local compression, stretching, and the optimal deformation of one of the two curves compared to the other, and the discrete wavelet transformation [29] ( ) measuring the dissimilarity between the wavelet approximations associated to the observations of curves.
For those based on data values and behaviour [28, 30] ( ) (5), a temporal correlation (6) between 2 functional data (curves), were combined with each of the dissimilarity measures based only on data values. The contribution of data value part and temporal correlation part were adjusted by an adaptative function according to a value of a given non-negative parameter (Table 1). The Table 1 comes from the original article [30] which developed the dissimilarity measure. So a dissimilarity measure based on data values and behaviour was obtain by combing one of the 4 measures based only on data value ( , , , ) and the temporal correlation (6) according for each value of the parameter in Table 1.
Thus, a total of 20 dissimilarity measures were used in this analysis. For example, the is the dissimilarity measure based on temporal correlation (CORT) and the Euclidean distance ( ) (equation 5) when ξ is equal to 1 (Table 1), this considered 46.2% of behavior contribution (given by CORT) and 53.7% of values contribution (given by ). The others dissimilarity measures were obtained in the same way and when =0 (Table 1) we had the 4 dissimilarity measures based only on values.”
Point 6: Line 115: Table 1: Are the numbers listed in Table 1 based on the data or are these
theoretical values? Further clarification is required here, in the text and/or caption for Table
- If these values are based on the real data, Table 1 would be better placed in the results.
Response 6: Table 1 is based on theoretical values from its original article research. We have added the reference in line 185-186 as “ The Table 1 comes from the original article [30] which developed the d_CORT dissimilarity measure.”
Point 7: Line 134: ‘Q patterns’. How was Q chosen?
Response 7: At this step we described the methodology, and so we had not yet the number of patterns (clusters). So, Q is the supposed number of cluster or patterns. To be clear we propose this modification in line 232 “Let Q be the number of patterns identified by the HAC, …”
Point 8: Line 184: Figure 3, Panel B. A brief description of codes in this plot would be helpful here
(e.g. 3FDA)
Response 8: Thank you, we better specified this in line 301-305, Legend Figure 3 as “4DTWCORT3 is the assessed HAC result with the potential number of patterns chosen 4, and performed with DTWCORT3 dissimilarity measure taking into account the 9.4% of d_dtw (data value) and 90.5 % CORT (data behaviour) (Table 1, xi=3), 3FDA is the assessed HAC result with the potential number of patterns chosen 3, and performed with d_FDA dissimilarity measure taking into account 100% of data value, etc.”
Additional comments by authors in the manuscripts:
We have added an interpretation of deceleration and acceleration situation through the velocity and acceleration function line 360-368 as “In both growth and decline phase of malaria incidence patterns, when velocity and acceleration functions had the same sign in an interval then malaria incidence patterns was in acceleration situation and when they had opposite signs malaria incidence patterns was in slowdown (deceleration) situation . For example, between the dates of the onset (A) and the slowdown (C) of seasonal outbreaks of patterns, velocity and acceleration functions were both positive, so malaria incidence patterns were increasing rapidly. Between the dates of slowdown (C) and the peak (D), velocity function was positive while acceleration function was negative, malaria incidence patterns were also increasing but slowly until achieving the peak.”
We have modified line 476-482 by "An in-depth analysis of their velocities and accelerations showed that the acceleration of phase decline (DE) was not direct on these seasonal epidemics, since they were disrupted by a small slowdown phase indoors. Indeed, during the DE phases of these 3 particular seasonal epidemics, there was exceptionally a moment when the velocity was negative and the acceleration positive (which translates into a slowdown), then the acceleration became negative again (still while the velocity was negative) to continue its phase of acceleration of the decline.”

Reviewer 3 Report
The authors applied functional data to identify patterns of malaria incidence in 575 villages in west-central Senegal from 2008 to 2012. Their modelling approach reveals the key three distinctive patterns of incidence degrees which may infer the strategic application of control measures. The manuscript is generally well-written and please see below for my comments.
Major comments:
- The model in equation (2) has basis functions. The authors use cubic B-splines to avoid periodic smoothing. However, considering the periodic nature of malaria incidence, using trigonometric functions will be a better basis function. With the choice of basis function affects the results or conclusion?
- The authors did not provide estimates for the coefficient in model (2). How does did coefficients differ by village and time?
- What is? It is not explicitly defined.
Minor Comments:
- Line 43 the comma before “would” should be removed
- Line 193-196, there is a reference to Figure 5, I think that should be Figure 4. Otherwise, the reference to figures should be in ascending order.
- Line 296, unnecessary new paragraph. Paragraph one and two under the discussion section should be one as the authors are trying to summarise their results.
Author Response
Point 1: The authors applied functional data to identify patterns of malaria incidence in 575 villages
in west-central Senegal from 2008 to 2012. Their modelling approach reveals the key three
distinctive patterns of incidence degrees which may infer the strategic application of control
measures. The manuscript is generally well-written and please see below for my comments
Response 1: Thank you for reviewing our manuscript.
Point 2: The model in equation (2) has basis functions. The authors use cubic B-splines to avoid
periodic smoothing. However, considering the periodic nature of malaria incidence, using
trigonometric functions will be a better basis function. With the choice of basis function
affects the results or conclusion?
Response 2: Thank you for this question. This aspect was discussed in the second paragraph of Discussion section line 296-300, this was the reason we had not explained deeply this choice. However, now we have added an explanation.
We agree the malaria incidence is periodic, but the level of intensity between season are not the same. This was the mainly reason we have not used the Fourier basis functions with a yearly periodicity (every 52 weeks example) because the result would give the same intensity level of incidence for each seasonal outbreak every year. However, we can use also Fourier basis by specifying that the periodicity is the whole studied period (all the 261 weeks), and in this case, the result didn’t change really compare to B-Spline, and this latter is more computationally tractable .
This sentence is added in line 153-156 as “In this work, we used cubic B-splines to avoid periodic smoothing [4, 5, 7]. Indeed, even if malaria incidence has a periodic nature, the level of intensity of incidence is not the same over seasons, in contrast with Fourier basis function which will show same level intensity of seasonal incidence.”
Point 3: The authors did not provide estimates for the coefficient in model (2). How does did
coefficients differ by village and time?
Response 3: The coefficients were determined by least squares method by minimizing the equation (3). This was mentioned under the equation (2).
The equation (2) showed the dependence between village, time, and coefficients by i indicator and time t. In equation (3) the smooth function of each village i, x_i(t_j), is replaced by the equation (2). Thus cik (coefficient) were estimated based on the observed data yij by least squares method.
This is clarified in line 138-140 as “… cik are the coefficients obtained by the least squares method by minimizing the penalized error sum of squares (3) after replacing in (3) the formula of x_i (t_j) by equation (2).”
Point 4: What is? It is not explicitly defined.
Response 4: Something is missing at the end of your question. We suppose this is P? We had forgotten to define it, we are sorry. P is the whole period interval with continuous time inside [1,261]. This is now mentioned in line 141-142 as “… P the studied period expressed here as [1, 261] in which times are continuous…”
Point 5: Line 43 the comma before “would” should be removed
Response 5: Thank you, the correction is done.
Point 6: Line 193-196, there is a reference to Figure 5, I think that should be Figure 4. Otherwise,
the reference to figures should be in ascending order.
Response 6: You are right. This paragraph and its Figure 5 are moved now bottom Table 3 in line 345-354.
Point 7: Line 296, unnecessary new paragraph. Paragraph one and two under the discussion
section should be one as the authors are trying to summarise their results.
Response 7: The paragraph 2 discussed the choice of B-spline at the place of another basis function, so this is not included in the summarized result. Since this is added in Method section because of your suggestion in point 2 we have removed this to avoid repetition.
Additional comments by authors in the manuscripts:
We have added an interpretation of deceleration and acceleration situation through the velocity and acceleration function line 360-368 as “In both growth and decline phase of malaria incidence patterns, when velocity and acceleration functions had the same sign in an interval then malaria incidence patterns was in acceleration situation and when they had opposite signs malaria incidence patterns was in slowdown (deceleration) situation . For example, between the dates of the onset (A) and the slowdown (C) of seasonal outbreaks of patterns, velocity and acceleration functions were both positive, so malaria incidence patterns were increasing rapidly. Between the dates of slowdown (C) and the peak (D), velocity function was positive while acceleration function was negative, malaria incidence patterns were also increasing but slowly until achieving the peak.”
We have modified line 476-482 by "An in-depth analysis of their velocities and accelerations showed that the acceleration of phase decline (DE) was not direct on these seasonal epidemics, since they were disrupted by a small slowdown phase indoors. Indeed, during the DE phases of these 3 particular seasonal epidemics, there was exceptionally a moment when the velocity was negative and the acceleration positive (which translates into a slowdown), then the acceleration became negative again (still while the velocity was negative) to continue its phase of acceleration of the decline.”
Round 2
Reviewer 1 Report
I feel that the authors have responded adequately to most of my comments.
The manuscript would benefit from careful editing of language.
Author Response
Response to Reviewer 1 Comments
Thank you for reviewing our manuscript.
We are pleased to learn that most of the comments have been adequately addressed.
We have followed your suggestion about the editing of language, thus a corrected version is submitted.
Best regards